# MRI Findings in Axial Psoriatic Spondylarthritis

**DOI:** 10.3390/diagnostics13071342

**Published:** 2023-04-04

**Authors:** Loredana Sabina Pascu, Nicolae Sârbu, Andrei Vlad Brădeanu, Daniela Jicman (Stan), Madalina Nicoleta Matei, Mihaela Ionela Sârbu, Doina Carina Voinescu, Aurel Nechita, Alin Laurențiu Tatu

**Affiliations:** 1“Sf. Ioan” Clinical Emergency Children Hospital, 800487 Galati, Romania; 2Faculty of Medicine and Pharmacy, “Dunarea de Jos” University of Galati, 800008 Galati, Romania; 3“Sf. Andrei” Emergency County Clinical Hospital, 177 Brailei st, 800578 Galati, Romania; 4Infectious Diseases Dermatology Department, “Sf. Parascheva” Infectious Diseases Clinical Hospital, 800179 Galati, Romania; 5Multidisciplinary Integrated Center of Dermatological Interface Research MIC-DIR, 800008 Galati, Romania

**Keywords:** psoriatic arthritis, magnetic resonance imaging, ankylosing spondylitis, skin biopsy

## Abstract

Psoriatic arthritis is a significant medical condition with a high prevalence, a wide variety of non-specific symptoms, and a high degree of overlap with other spondylarthritis disorders, particularly ankylosing spondylitis. Hence, knowledge of the magnetic resonance imaging (MRI) manifestations and a multidisciplinary strategy are required for the better management of these patients. We searched publications from the last 10 years and focused on the most relevant ones which discussed the classification criteria, the MRI characteristics of axial psoriatic arthritis, the importance of MRI for follow up, and the reliability of skin and synovial biopsy. Axial spondylarthritis can be diagnosed and followed up on using the well-established MRI technique and, additionally, a biopsy. The analysis and concordance between them can provide new directions for future studies.

## 1. Introduction

Spondylarthritis (SpA) represents a group of rheumatic diseases that include ankylosing spondylitis (AS), axial and peripheral spondylarthritis, psoriatic arthritis (PsA), inflammatory bowel disease-associated spondylarthritis (IBDSpA), reactive arthritis and, non-radiographic axial SpA. They all have some standard features, including pain, stiffness, axial involvement, arthritis, dactylitis, enthesitis, and a high prevalence of the HLA-B27 allele. There are multiple terms to indicate patients with axial PsA, such as psoriatic spondylitis (PS), axial psoriatic arthritis (axPsA), or psoriatic spondyloarthritis [1].

Initially, PsA was considered part of rheumatoid arthritis (RA) until psoriasis (PsO) was linked to aggressive forms of inflammatory arthritis that had a different articular pattern from RA, affecting the distal interphalangeal joints and the sacroiliac joints. After discovering the rheumatoid factor, they were classified into two different entities. Even so, the rheumatoid factor is not specific to RA, and a false-positive test might mislead the diagnosis of PsA [2]. It is a potentially debilitating disease, affecting 0.4 to 2% of the general population and up to 19.7% of patients with skin or scalp psoriasis. It has a higher prevalence in Europe—22.7%—with the mean age of individuals with axial PsA being around 51 years old [3]. It does not have a male or female predilection, compared with other arthropathies. On the other hand, axPsA symptoms can develop in older patients, unlike for AS. SpA develops at a younger age and is more prevalent in males who are HLA-B27-positive [4,5,6,7]. There are two phenotypes of PsO: type I (85% of patients) is characterized by the beginning of symptoms in young adults before the age of 40, often with a severe evolution; and type 2 (15%) beginning in patients after the age of 40 with a milder evolution [8].

A multicenter European study involving 1560 subjects in 2006 showed that 31% of patients with psoriasis might develop PsA after 30 years, with the prevalence increasing over time [9], unlike the pediatric population where arthritis occurs prior to psoriatic skin lesions up to 10 years, typically in children before the age of 10, while only 2% of children with psoriasis will encounter juvenile idiopathic arthritis (JIA) [10,11]. Psoriatic arthritis is an immune-mediated, inflammatory, multifactorial disorder in genetically susceptible patients. In a genetically predisposed individual, factors such as obesity, infections, articular trauma (sprain or dislocation), or skin trauma (triggering the Koebner phenomenon) can lead to a self-sustained immune response and clinical manifestations [12]. The spondyloarthropathies are characterized by articular and periarticular inflammatory modifications, with a destructive evolution, different from mechanical changes in non-inflammatory conditions [13].

Radiological involvement can be observed late in the evolution of inflammatory changes, so MRI evaluation has gained more and more ground in recent years in the early diagnosis of PsA. However, there needs to be more international consensus regarding the definition of axPsA [14].

In recent years, papers concerning PsA and its extra-articular manifestations have been published, but the features of axPsA are still a matter for debate. Most axPsA patients have associated peripheral articular involvement, which has been widely studied. However, axPsA is still challenging to diagnose due to its unspecific symptoms and biological findings [15].

Collaboration between dermatologists and rheumatologists is essential, as they have an important role in diagnosing PsA, since psoriasis might precede arthritis by 7–12 years [16]. Screening patients with psoriasis without initial articular rheumatic features and identifying the early signs is fundamental for preventing irreversible evolution, joint damage, and function loss.

MRI allows for the assessment of morphological details in axial and peripheral articulations without ionizing radiation. Its main limitations are high costs and the fact that it is a time-consuming procedure [16].

We reviewed the literature from the past 10 years for publications about MRI findings in axPsA. MRI is a well-established technique for diagnosis and follow-up in axial spondylarthritis but has not been studied extensively in axial PsA.

## 2. Materials and Methods

We performed searches from January 2013 to May 2022 to identify articles referring to MRI findings in axial PsA, as detailed in Figure 1. We used keywords such as “axial psoriatic arthritis,” “MR”, “MRI,” “spondylarthritis”, “biopsy”, “psoriasis”, “psoriasiform”, “sacroiliitis”, “inflammatory back pain”, “juvenile idiopathic arthritis”, “juvenile idiopathic psoriatic arthritis”, “dactylitis”, “treatment”, “side effects”, and “DMARDs” in the PubMed, ScienceDirect, and SpringerLink electronic databases. We selected 96 studies and reports that we have cited in our review. Among these, a list of the most relevant articles is shown in Table 1.

## 3. Results

### 3.1. Clinical Manifestations

PsA might be a challenge in terms of diagnosis, mainly because of the absence of any specific biomarker. Patients often complain of long-standing back pain with an insidious onset before any other symptoms. The rheumatologist needs to differentiate between inflammatory back pain (IBP) and mechanical back pain (MBP), as both of them may occur in axPsA patients and their accurate differentiation allows for appropriate treatment administration [30,37] (Table 2). Patients with IBP usually complain of lower back stiffness in the morning for at least 30 min, with no improvement at rest, sometimes waking patients at night or early in the morning, with an improvement in symptoms after starting physical activity and a good response to non-steroidal anti-inflammatory drugs (NSAIDs) [15]. Typically, IBP last over 3 months, with a gradual onset in middle-aged adults, under 40 years old; however, studies have revealed that 45–55% of axial PsA patients do not report IBP symptoms [30]. In contrast, MBP may occur at any age, typically over 50 years old, with an improvement in symptoms at rest and worsening with movement, and an insidious/acute onset.

**Table 2 diagnostics-13-01342-t002:** Differences between IBP and MBP.

Pain Features	IBP	MBP
**Age**	<40 yo	>50 yo
**Onset**	Insidious	Variable
**Timing**	During the night, can wake the patient.	Daytime
**Improvement**	With physical activity, no improvement at rest.	With rest, movement may worsen the pain.
**Diagnosis**	NO history of back injury or heavy work load; Alternating buttock pain; Favorable response to NSAIDs.	History of back injury or heavy work load; No or minimal response to NSAIDs *.
**Duration**	>3 months	<4 weeks; persisting pain needs additional tests and imaging (HLA-B27, acute phase reactants, MRI, clinical examination).
**Location**	Anywhere; the proximal/distal third or the posterior region are highly suggestive.	Anterior mid-third of sacroiliac joints, often bilateral.
**Treatment**	See Figure 2.	Physiotherapy, NSAIDs, corticosteroids, when necessary.

* NSAIDs, nonsteroidal anti-inflammatory drugs.

**Figure 2 diagnostics-13-01342-f002:**
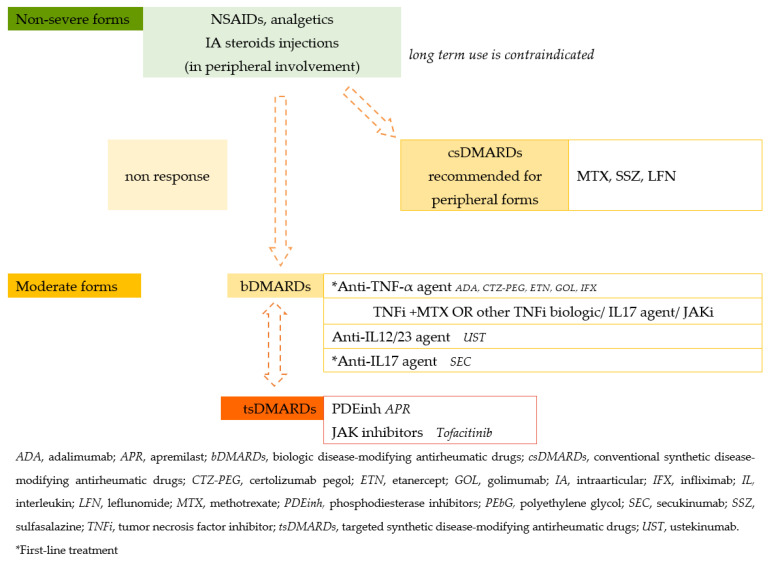
Treatment guidelines for PsA.

Nonetheless, IBP cannot be used as a specific criterion since it is typical of other subsets of SpA and even more frequent in AS patients [4]. AS patients report more frequent and intense spinal pain, while axPsA is associated more frequently with dactylitis, enthesitis and peripheral arthritis, and worse peripheral destruction, and only 4% of the patients have pure axial disease [25,28,38]. Additionally, Abdelaziz et al. revealed that psoriatic nail dystrophy and dactylitis are exclusively seen in axPsA patients, which may help in the differentiation of these two entities [28]. Most of the classification criteria use clinical, genetic, laboratory, and imagistic information, but many signs and symptoms are not specific to a particular subset of SpA. The classifications have a sensitivity and specificity that do not generally exceed 80% [39]. They define a homogenous group of patients that are easier to study [40].

MRI turned out to be the cornerstone in the early detection of minor signal modifications, such as bone marrow edema or fatty lesions in the spine and sacroiliac joints (SIJ), to estimate the probability of a suspected disease in patients with specific symptoms. Given the overlaps between PsA, SpA, and AS, currently, the most used classification for the SpA group is Assessment of SpondyloArthritis international Society (ASAS) classification (98.4% sensitivity, 77% sensitivity), which includes both MRI characteristics for diagnosis, and the modified New York classification [41]. The ASAS criteria are often used as diagnostic criteria in both research and clinical practice [39].

A small percentage of patients who present with isolated spine inflammatory changes, with negative HLA-B27 and no evidence of sacroiliitis on MRI, will not meet the ASAS criteria [42]. Fortunately, the ASAS and the North American Spondyloarthritis Research and Treatment Network (SPARTAN) have recently started a new study that aims to improve ASAS criteria specificity [43].

Today’s most widely accepted and applied classification criteria for PsA is the CASPAR criteria set (Classification of Psoriatic Arthritis), which includes dermatological, clinical, and radiological criteria [40,44], developed for a homogeneous group of patients in clinical trials and providing guidance for practitioners, with a high sensitivity of 91.4% [40,44] and 98.7% specificity [2]. Even though CASPAR criteria is a little less sensitive than the Vasey and Espinoza criteria, which include the family history and is applied to both RF-negative and -positive patients, it presents better clinical applicability over other criteria sets, such as Vasey and Espinoza or Moll and Wright criteria [38,39]. Recent studies have revealed that the CASPAR classification is more sensitive in terms of the diagnostic classification assessment of both early and late PsA disease compared to the ASAS criteria [45]. The Brazilian Society of Rheumatology 2020 guidelines for psoriatic arthritis recommend clinical and imaging criteria for diagnosis and the CASPAR criteria for disease classification [46]. Still, it has limitations in cases of recent onset (Table 3).

On the other hand, for the pediatric population with juvenile arthritis, the most common classification system is that of the International League of Associations of Rheumatology (ILAR). ILAR classifies JIA into seven subgroups:Systemic arthritis;Olygoarthritis (persistent/extended);Polyarthritis RF-negative;Polyarthritis RF-positive;Psoriatic arthritis;Enthesitis-related arthritis (ERA);Undifferentiated arthritis (UA).

Despite the differences between adult and juvenile PsA (jPsA), the CASPAR criteria set may diagnose more JIA patients with juvenile PsA [33], therefore it may be useful for further revisions and a more specific classification.

The COVID-19 pandemic improved access to remote medical consultations through telemedicine. Patients could benefit from consultations, a diagnosis of PsA, and faster treatment. The better detection of PsA was observed through telemedicine, since it helped clinicians use proper diagnostic criteria and screening tools and better understand the characteristics of PsA. Screening tools via telemedicine included validated ones, such as Psoriasis Epidemiology Screening Tool (PEST), and non-validated ones, such as mnemonic PSA (Pain, Stiffness/Sausage finger, and Axial spine involvement) [48].

Post-pandemic studies have revealed that some patients developed chronic arthritis, including psoriatic SpA with bilateral sacroiliitis modifications, validated on MRI, after COVID-19 infection, in the absence of a family history of PsO or HLA-B27 [49,50]. In addition, a paper published in 2022 [51] claimed that axial PsA might undergo exacerbation after the COVID-19 vaccination.

### 3.2. Imaging Protocols

The primary imaging tool widely used in diagnosing inflammatory joint disease has been conventional radiography, included in the modified New York criteria. Its sensitivity is low for early disease. Nevertheless, it remains a valuable investigation tool in evaluating structural disease progression and is widely available, fast, cost-effective, and requires a low dose of ionizing radiations compared to computer tomography (CT) [52].

However, MRI is the best imaging tool for assessing preclinical PsA, identifying and monitoring inflammatory and structural damage in terms of both axial and peripheral changes, and facilitating better visualization of the peculiar changes involved in PsA. Some studies claim ultrasonography is more sensitive in evaluating peripheral PsA (metacarpophalangeal—MCP, proximal interphalangeal—PIP, and distal interphalangeal—DIP joints) [2,53,54,55] compared to radiography or MRI. In contrast, other recent studies assert that MRI is more confident in detecting morphological details, but is more expensive, time-consuming, not widely available, and might require contrast agents [52]. In axial disease, the best imaging modality for both inflammatory (bone marrow edema (BME), osteitis) and structural damage, such as erosions or fat deposits, remains MRI investigation [52]. Nevertheless, MRI has limitations in characterizing bone remodeling, the morphology of syndesmophytes, or appreciating sclerosis [15], and it is contraindicated in patients with pacemakers or ferromagnetic metal implants [16].

The MRI examination protocol includes the following steps [41,56,57].

#### 3.2.1. Axial PsA

-Sagittal T1-weighted (T1w) sequences—best for the characterization of fat content or to assess structural bone lesions.-Sagittal T2-weighted Fat-Sat fast spin echo sequences and short tau inversion recovery (STIR) sequences (short tau inversion recovery) sequences in two planes for water content evaluation/BMO.-For a better assessment of the costovertebral and costotransverse joints and the facet joints, coronal sequences can be added to the protocol [58].-Sagittal T1w Fat-Sat sequences with gadolinium enhancement—used for the detection of osteitis; increased perfusion; rarely used, in cases of doubt and high suspicion, to differentiate exudate from synovitis or to assess the activity of bone erosions [59].

#### 3.2.2. Sacroiliac Joints Sequences

-T1-weighted axial oblique and semi-coronal sequences.-T1-weighted Fat-Sat (TIWFS) spin echo/T1 Dixon/3D gradient echo such as VIBE—for the better evaluation of erosions; VIBE sequences showed a better ability to detect cartilage erosions before extending to the underlying bone [60].-Axial oblique semi-coronal STIR and T2-weighted fat-suppressed (T2WFS) sequences (coronal plane tilted parallel to the long axis of the sacroiliac joint) with 4-mm slice thickness.-Apparent diffusion coefficient (ADC) map—STIR or T2WFS sequences may be substituted or supplemented for the better assessment of SIJ inflammation.-Coronal and axial oblique T1w Fat-Sat with gadolinium—detects osteitis; increased perfusion; recommended in cases of doubt and high suspicion; T1WFS pre- and postcontrast administration can differentiate active inflammation from a simple fluid.

Intravenous contrast can be omitted if it is aimed at detecting bone erosions, edema, and bone productions. However, in active PsA, in addition to bone edema, post-contrast sequences show periosteal enhancement and synovitis, and differentiate active bone erosions from remission [59]. Similar MRI protocols are used in spondylarthritis for diagnosis and follow-up.

### 3.3. MRI Findings

MRI evaluation of the spine and sacroiliac joints indicates the location of lesions and the distinction between the cartilaginous and ligamentous joint compartments. It can assess bone marrow edema and structural lesions, such as sclerosis, squaring, erosion, spine syndesmophytes/joint space width, or the ankylosis of sacroiliac joints [59]. 

However, there needs to be a universal consensus regarding MRI scores. There is a broad range of studies, including classifications such as the Berlin modification ankylosing arthritis spine MRI activity score, and the Ankylosing Spondylitis Spine Magnetic Resonance Imaging-activity (ASspiMRI-a), Spondylarthritis Research Consortium of Canada (SPARCC), and Psoriatic Arthritis Magnetic Resonance Image scoring system (PsAMRIS), to assess disease activity in spondylarthritis group diseases [61,62]. Based on a multireader experiment, it is difficult to select one of the three methods for inflammation assessing of the spine and SIJs. The ASspiMRI-a and Berlin methods offer a better overall picture of spine inflammation; however, the SPARCC approach may have advantages in terms of reliability due to its greater consistency in this area [61]. Two large working groups from Canada and Denmark developed and validated an MRI scoring system called CANDEM for better evaluating inflammatory and structural lesions over time, as well as the treatment response. Unfortunately, it has limited clinical applicability since it requires a particular acquisition protocol with 4-mm-thickness T1-weighted turbo spin-echo and STIR sequences on the sagittal plane. Separating the cervical, thoracic, and lumbar spine acquisitions with central and lateral sagittal sequences is helpful for a better assessment of the posterior elements of the spine [15]. This scoring method may be useful in an evaluation of how various medications affect the particular elements of inflammation and damage in the spine, the relationships between lesions, and the progression of inflammation and damage throughout the entire spine [63].

Axial psoriatic arthritis includes the following findings [57].

#### 3.3.1. Vertebral Findings (Similar to Spa Findings)

Inflammation of the vertebral body superior or inferior corners, identified in the early stages as a low signal in T1w, a high signal in T2w and STIR, due to bone marrow edema and later, a high signal in T1w and T2w due to fatty bone marrow degeneration, known as the “shiny corner sign” or “Romanus lesions” (Figure 3). This is also found in ankylosing spondylitis.Spondylodiscitis—inflammation of the whole vertebral end plate, involving the adjacent intervertebral disc (Andersson lesion) and soft tissue, identified on low signal T1w in both the intervertebral disc and adjacent end plates due to inflammation and bone marrow edema; on high signal T2w Fat-Sat and STIR in the disc space, adjacent endplates, and paravertebral soft tissue, also involving the psoas muscle, with the loss of the endplate cortex signal. Normally, muscle tissue has an intermediary signal in both T1w and T2w sequences, while inflammation leads to an increased signal in both sequences. A T1w-sequence with gadolinium shows enhancement of the vertebral endplates and paravertebral soft tissue and peripheral enhancement in the case of collections. DWI sequences distinguish the acute and chronic stages (high signal vs. low signal). It is described also as a complication in the evolution of advanced SpA, mostly in the lumbar spine [64,65].Facet joint inflammation—generated mostly by articular degeneration, usually associated with BME within the spinal pedicles, and common in most arthropathies.Inflammation of the posterior and lateral elements—including the costovertebral joints, transverse and spinous processes, and the adjacent soft tissueBone erosion—best visualized on T1-weighted sequences as cortical (dark appearance) bone defects, contour irregularities, and the loss of the normal bright appearance of the adjacent bone marrow.Bulky new bone formation—bone productions in high signal T1w; marginal and paramarginal vertical syndesmophytes distributed asymmetrically along the spine, with a peculiar feature of late PsA, while in SpA, syndesmophytes are typically bilateral and symmetrical, with only a marginal distribution and evolution from caudal to cranial [66]. In axial PsA, syndesmophytes extend towards the adjacent vertebra, while in AS, they are continuous from adjacent vertebra, with a tendency to the formation of osteophytic bridges and further evolution to the bamboo spine [67]. The vertebral joint spaces are preserved until the late stages of the disease.Enthesitis—involving the supraspinal ligament, interspinal ligaments, and ligamentum flavum; normally, tendons have a homogenous low signal in all sequences [65].

#### 3.3.2. Sacroiliac Joints Findings

BME is evident in low signal T1w, hyper signal T2w, STIR images and +C T1w Fat-Sat, similar to blood vessels and spinal fluid; the signal intensity is directly proportional to the inflammation activity. It is usually located periarticular to or on the subchondral bone surfaces and it is an indicator of disease activity (Figure 4) [65].Capsular inflammation—increased signal on T2w GRE and STIR sequences.Enthesitis—entheses are normally seen as hypointense structures, whereas inflammation leads to an increase in signal; best visualized on T2w Fat-Sat and STIR sequences. There are similar findings in both PsA and SpA [65].Joint space fluid.Joint space enhancement.Erosions–initially focal, later they will converge and will have a pseudo-enlargement aspect of the sacroiliac joint. In T1w images there is a loss of cortical bone signal (normally hypointense) and bone marrow fat (normally hyperintense).Inflammation at the site of erosion.Fat metaplasia in an erosion cavity or ‘backfill’ [65].Sclerosis—better visualized on an X-ray or CT scan; a subchondral or periarticular area with a low signal compared to normal bone marrow on T1, T1FS (SPIR), and STIR sequences (Figure 5).Slight: <25% of the subcortical bone area. Moderate: 25% to <50% of the subcortical bone area.Severe: >50% of the subcortical bone area [29].Ankylosis [65]Partial: Partial osseous bridging across the joint space.Total: Fusion of the joint facets [29].Bone bud—new bone products that are not bridging the joint [68]. In axial PsA syndesmophytes extend into the SIJ space, while in AS, they are across the SIJ space [67].

The active inflammatory changes in PsA assessed by MRI include subchondral bone edema, enthesitis, capsulitis, synovitis, and enhancement inside or adjacent to the joint. Chronic inflammatory damage on MRI consists of periarticular erosions, fat metaplasia, subchondral bone sclerosis, joint space narrowing, syndesmophytes, and ankylosis [17,20,57]. 

Recent studies have shown conflicting results in terms of axial PsA assessment. MRI findings in PsA are less severe than in AxSpA, with lower scores for bone marrow edema at the vertebral and sacroiliac joints. Furthermore, there is a negative correlation with HLA-B27, which makes it challenging to develop a classification system [4]. Some studies have indicated that HLA-B27 positivity in axial forms of the disease is associated with high susceptibility and severity, including more extensive and severe inflammatory changes [7,15,18,20]. 

Poddubnyy et al. observed that up to 44% of PsA patients were HLA-B27-positive, compared to 90% of axial SpA patients [38]. Meanwhile, in a study by Diaz et al. [21], only 10% of PsA patients were HLA-B27-positive. 

PsA is a heterogeneous disease due to its frequent spine and sacroiliac joint involvement. The axial involvement can be extensive, with diffuse vertebral involvement associated with bulky vertical pseudosyndesmophytes that emerge from the spinal ligament [65,66], unlike in AS, where lesions are well defined and rarely associated with vertebral para marginal/“non-marginal bulky” syndesmophytes [5] (see Table 3).

Dorsal spine involvement in AS is characterized by higher vertebral osteoproliferative severity with symmetrical, marginal syndesmophytes [25], unlike in axPsA, where paravertebral ossification is uncommon, and bone productions are non-marginal and less symmetrical. Fusion facet joints were frequently found in AS lumbar spines, while PsA syndesmophytes may occur without sacroiliitis. Among patients with axPsA there was less severe asymmetrical sacroiliitis and less frequent IBP [5,18]. 

AS often involves important joint damage from the sacroiliac level evolving from caudal to cranial with the apophyseal joints’ involvement and the formation of bulky syndesmophytes, while in axPsA inflammation was observed more frequently in the cervical spine (frequent fusion of facet joints), with minimal apophyseal joints involvement and less bone production [4,5], even in the absence of sacroiliitis or other significant spine damage in early disease [30,41]. 

On the other hand, Gazel et al. observed that active changes in the cervical spine are more frequent in AS patients than in PsA patients but with a similar rate of structural change. Compared to AS, PsA patients had numerically fewer changes in the cervical spine without reaching statistical significance [69]. 

Up to 75% of axial PsA patients with longstanding disease have cervical spine damage involving C1-C2, where odontoid erosions and atlantoaxial instability can be found, leading to further complications, such as spinal cord compression and neurological deficit [1,15]. The lower cervical spine might undergo bone production, inflammation of the inter-apophyseal joints, and ossification of the anterior longitudinal ligament [15]. 

According to Jadon et al., one out of four patients with PsA and AS fulfill the classification criteria for both diseases. The term psoriatic SpA is used for cases that share features of both entities [25]. Axial PsA patients seem to have similar sex predominance, disease activity, and functionality, but PsA patients are older at diagnosis time, with a median age of 49 years old, compared to 43 years old for AS patients, with much more smokers in the PSA group [27]. Moreover, SpA is genetically and immunologically linked to inflammatory bowel disease (IBD); thus, IBD may predict arthritis and two out of three SpA patients have subclinical IBD, with similar changes as those encountered in Crohn’s disease [70]. IBD has a similar incidence in axial PsA and SpA and a higher incidence in pediatric patients with jPsA or psoriasis-associated conditions in JIA [11,27].

On the other hand, in a study that analyzed each vertebral level of structural damage, men with AS had higher score levels than women. The C5 and L4 vertebrae were more involved than others [71,72]. 

A study on 402 PsA and AS patients concluded that both have similar activity, metrology, and disability [25]. Similar findings were confirmed in another study published in 2021, which showed that both active and structural changes in PsA and AS patients had similar rates in the lumbar spine and SIJ and were less frequent in the cervical spine. In the thoracic spine more active and structural damage in PsA patients was recorded, but without reaching statistical significance [69] (Table 4). 

The anterior chest wall (ACW) is usually overlooked while examining the extension of axial PsA. Research conducted between 2016 and 2019 using standard clinical practice methods examined 104 individuals with PsA, 45 of whom had the axial illness. Sacroiliitis affected 71% of the axial PsA patients. The most frequent bone marrow edema was identified in the sternoclavicular joints, followed by the manubriosternal and costosternal joints. MRI sequences showed bone marrow edema, synovial hyperemia, capsular structure thickness, erosions, bone irregularities, bone productions, and osteophytes [73]. 

Recent studies have shown that many patients might have active spine changes without sacroiliitis [21,25,69]. In a study published in 2022, 91.7% of cases of MRI-spondylitis were in confirmed PsA patients, while 8.3% of cases were in patients with psoriasis and axial SpA [21]. 

Even though ALs are exceptionally identified at the disease onset, up to 6% of PsA and AS patients might encounter them. There were a few cases reported where symptomatic patients who were HLA-B27-negative did not meet the ASAS criteria for ax SpA but were identified on MRI scans with erosive disco-vertebral lesions without sacroiliitis modification. Nevertheless, these patients met the CASPAR and GRAPPA criteria for axial PsA [74]. 

Patients with PsA may have inflammatory or mechanical back pain, but may also have axial disease on imaging while experiencing no back pain or late pain onset [41], since over 30% of axPsA patients may not have IBP. Withal, IBP may be present in patients without axial PsA, so IBP presents a low correlation with imaging findings of sacroiliitis or spondylitis [30]. On the other hand, clinical complaints such as morning stiffness and pain during the night are highly associated with an ASAS MRI-positive and MRI-global impression [32]. MRI changes, such as subchondral BME (Figure 4), might be expected in both PsA patients and healthy people, especially recreational runners, professional ice hockey players, or women in the postpartum period (who present mechanic SIJ modification), leading to significant “false positive” diagnoses of PsA. However, sacroiliac joint inflammation does not exceed three quadrants in these cases, unlike in the SpA group [13,22,23,43,75,76].

In Gazel et al.’s study, only 16% of patients had active inflammation at both sites [69]. When small, well-defined, abnormal bone marrow signal intensities are seen on the SIJ, such as edema or sclerosis found on one, two, or, far less often, three and four consecutive slices, with no severe associated erosion, they are irrelevant. These MRI findings are better seen on coronal oblique images oriented on the S1–S3 axis, often in asymptomatic patients, or they might be associated with back pain. Diaz et al.’s study [21] showed that inflammatory changes in the spine and SIJ are less frequent compared to the high prevalence of back pain. At the same time, the axSpA and IBP criteria had poor sensitivity for MRI findings [1]. 

Mimics of sacroiliitis risk the over-diagnosis of inflammatory sacroiliitis with a subsequent negative impact on the patient’s lifestyle. The most frequent mimickers are stress-related changes, infection, osteoarthritis, stress/insufficiency fractures, and osteitis condensans ilii [67].

Mechanical stress loading is concentrated in the anterior mid-third of the SIJ and is typically bilateral and triangular shaped. BME also occurs in healthy, physically active individuals at the antero-posterior inferior quadrants, probably due to the different distribution of mechanical forces [37,77], while degenerative changes, most commonly bone sclerosis, are found at the anterior and middle thirds of the joint and at the antero-superior quadrant. Table 5 summarizes the main pathologies of the differential diagnosis of BME, based on the typical locations. BME or sclerosis noted on the proximal/distal third of the posterior quadrant of the joint along with structural lesions, such as bone erosions or backfill, are strongly suggestive of sacroiliitis. [13,22,76,78]. The incidence of 30–41% of physically active individuals [78] that meet the ASAS criteria for axSpA advocates the importance of contextual MRI interpretation with a patient’s clinical and serological profile.

Insufficiency fractures are typically seen in osteoporotic elderly people, more often post-menopausal women, but are also seen in cases of steroid-induced osteopenia, infiltrative disease, and a history of pelvic radiation, demonstrated by the sacral alae and bilateral predominance [78].

Stress fractures are typically more common in athletes, shown on MRI as a unilateral BME on the sacral side, with no involvement of the subchondral bone; a vertical fracture line should be seen. Young athletes may be affected by vertebral pedicles and pars interarticularis fractures, leading to spondylolysis [78,79]. *Osteitis Condensans Ilii* is another condition that needs to be excluded when SIJs are evaluated; it is characterized by extensive subchondral sclerosis, frequently involving the ilium and very little BME or fat replacement, without significant erosions of the articular surfaces, unlike sacroiliitis, with a similar location to mechanical etiology changes, in the anterior part of the mid-third of the SIJ, often bilateral and well-defined with a triangular shape, but more extensive sclerotic changes [22]. BME has a ventral-cartilaginous joint segment of the ilium pattern and extends beneath the arcuate line, unlike axSpA which is often located at the dorsal-cartilaginous segment of the SIJ and rarely exceeds the arcuate line [80].

Sibel Zehra Aydin et al. performed a study on 1195 patients with axial PsA where they analyzed the sensitivity of the ASAS IBP criteria. They observed that inflammatory back pain criteria are limited in axial PsA diagnosis, with moderate sensitivity and a lower diagnosis rate in up to half of women. In addition, they noted that the Calin criteria had better agreement with the imagistic investigation than other criteria [81]. 

A study published in 2013 [81] showed that the extent of lesions in axial PsA, non-radiologic axial SpA (nr-axSpA), and AS patients was similar. However, the scores for the number of severe lesions in SIJ and MRI were lower in axial PsA and nr-axSpA than AS, using the semi-quantitative Leeds Scoring System for BME lesions. In addition, they observed that most of the PsA patients who were HLA-B27-negative had normal MRI scans, while the HLA-B27-positive patients had a similar pattern of BME as AS patients. So, HLA-B27 positive defines a group with more severe lumbar spine and SIJ lesions [81]. 

Shan-Shan Li et al. selected 186 patients who met the CASPAR criteria for axPsA and associated dactylitis. They noticed that these patients had higher disease activity and more severe joint damage. On the other hand, dactylitic axPsA patients had inflammatory back pain in 50% of cases. Withal, dactylitis is more common in axial PsA, than in peripheral disease [76,82]. 

Conventional, segmental MRI investigations limit the ability to assess psoriatic arthritis or spondyloarthropathy evolution, as only up to 5% of patients have pure axial disease [1,5]. Therefore, a whole-body MRI has been introduced, especially for patients who have both spine and sacroiliac joint damage. It is essential to fully assess the disease extent and activity, since peripheral damage often precedes axial modifications and can also be evidence of silent skeletal lesions [7,68]. Thus, up to 70% of peripheral damage leads to axial changes during the evolution process. In the early stages, only up to 28% of patients are at risk of simultaneous axial damage [7]. 

Following standard protocols, advanced techniques have developed improved MRI imaging, which includes quantitative MRI sequences, such as dynamic contrast-enhanced MRI (DCE-MRI), diffusion-weighted imaging (DWI), and chemical shift-encoded MRI (CSE-MRI) [68].

-DWI has the advantage of not necessitating gadolinium administration and is helpful in the early detection of spinal cord damage. ADC discriminates between active-inactive juvenile inflammatory arthritis, a better diagnosis of SpA, and mechanical versus inflammatory back pain.-CSE-MRI helps detect active inflammation and structural damage.-DCE-MRI is useful in the evaluation of inflammation activity, detecting early disease from periarticular soft tissue inflammation, even in the absence of synovitis [81]. It is also helpful in distinguishing synovitis between RA and PsA.

New techniques have been developed for better cartilage assessment [68]:
-T2- and T1p-mapping—providing insight into early biochemical cartilage changes and evidence of atlantoaxial instability, and does not require contrast agent administration.-DGEMRIC (delayed gadolinium-enhanced MRI of cartilage)—recommended for peripheral disease; useful in uncovering early biochemical cartilage disturbances; requires long acquisition protocols.

A 3T whole-body MRI study performed on 50 patients with psoriatic arthritis, spondylarthritis, or healthy subjects in 2021 revealed that 54 clinical enthesitis were observed from a total of 450 sites; the most common inflammation was detected in the anterior chest wall, involving the costo-sternal joints, sternal synchondrosis, and sternoclavicular joints in 44% of cases [83]. 

The IDEAL sequence provides excellent homogeneous fat suppression in areas of high magnetic susceptibility, delivering a better fat distribution map, helping in the quantitative assessment of the active sacroiliitis, and providing a more confident diagnosis than previous standard Fat-suppression sequences [84,85]. 

Even though axial PsA has some clinical and imaging particularities compared to AS, some studies still debate whether axial PsA is an independent disease or an axSpA with associated psoriasis [1]. 

A survey across 13 European countries between 2017 and 2018 found that patients visit 2.6 healthcare professionals before diagnosis, and there is a mean of 7.4 years’ delay between the first visit to the doctor and the time of diagnosis [86]. Similar findings were noted in the Benavent et al. study [18]. 

Other studies have shown that an additional spine MRI scan should be recommended when axPsA is clinically suspected, and spine screening does not add value to SIJ MRI scans since there is a low level of confidence regarding active inflammation in the spine [21,69]. 

However, whole spine MRI allows for simultaneous axial and peripheral evaluation and should be considered for further studies to develop better diagnostic criteria and definition, a better understanding of treatment responses, and the timely use of biological agents [21,24,52,62]. 

The European League Against Rheumatism (EULAR) 2015 MRI recommendations include monitoring the disease activity, assessing structural changes, and predicting the outcome and treatment effects [87]. 

#### 3.3.3. Specific Features of Juvenile Psoriatic Arthritis [88]

MRI findings in JIA include:-Early disease—periarticular osteopenia, effusion, juxta-synovial soft-tissue swelling;-Intermediate disease—narrowing of the spaces between joints, cortical erosions, epiphyseal overgrowth;-Late disease—ankylosis, joint angular deformities, contractures, muscle atrophy.

However, it is questionable whether MRI-detected bony depressions may be regarded as an outcome metric to indicate structural damage in JIA. Children have incomplete ossification and the subchondral bony contour may appear uneven and fractured, so it can be difficult to differentiate a normal from a pathological appearance, but thinner cartilage may indicate JIA [89].

JPsA has two age-peak incidences with distinct features [34]:1.The 2–5 years age group displays the phenotypic and pathophysiological features of common autoimmune diseases, including:
-Female predominance, positive anti-nuclear antibodies (ANA), higher predisposition of chronic uveitis.-Similar features to oligo- and polyarticular JIA or early-onset ANA-positive JIA.-Arthritis, often involving the knee and ankle; the involvement of dactylitis and distal interphalangeal joints is highly suggestive of jPsA. Dactylitis is present in 20–40% of patients and is the first musculoskeletal finding at presentation in about 15% of cases; moreover, it has been observed a long time before skin psoriasis [70].-Higher incidence of small joint and wrist involvement.-Enthesitis in 22% of cases, typically at the Achilles tendon and plantar fascia insertions into the calcaneus, but many patients who encounter both arthritis and enthesitis are classified as ERA, according to the ILAR criteria [90,91]. It is less frequently encountered than in late-onset jPsA.
2.The 9–12 years age group have features of autoinflammation that emerge as enthesopathy, including:
-Relative equal sex distribution, with a little male predisposition;-Resembles adult PsA features;-Enthesitis in up to 60% of patients [91];-Axial involvement, sacroiliitis;-HLA-B27 positivity, but some of these cases will be diagnosed as ERA or UA by the ILAR criteria.

Psoriasis may occur in jPsA with a delay that can reach 10 years [34]; simultaneous sacroiliitis or enthesitis along with psoriatic or psoriatic-like skin lesions exclude patients from a diagnosis of jPsA and ERA, which represents a hindrance in the evaluation of patients. Psoriasis in young children is often more subtle and more similar to atopic eczema or erythema and scaling behind the ears [10,34].

Oligoarticular peripheral onset is seen in both juvenile and adult PsA, where juvenile evolution tends to be polyarticular and less severe in terms of bone erosions and deformities. In addition, psoriasis is associated with a more severe prognosis [10].

Altered gut and skin microbiomes seem to be associated with PsA, albeit gut dysbiosis (with the loss of microbial diversity) may or may not present gastrointestinal symptoms, hence colitis at baseline may predict a chronic course of arthritis. This association is more difficult to see in pediatric patients, since gut inflammation is best identified by endoscopy, but has limited indications in pediatrics due to the invasive procedure risks and the requirement for general anesthesia. Hence, fecal protectin (fCAL) showed a high correlation with endoscopic results and a further correlation with sacroiliac joint inflammation in both adults and children [35,70]. Strikingly, the ILAR classification subdivides patients with onset arthritis at pediatric age into groups with different names to those in the adult classification [92], even if some of them have counterparts in the adult categories, for example:-Systemic JIA corresponds to adult-onset Still’s disease;-RF-positive polyarticular JIA is equivalent to RF-positive RA;-ERA has adult-equivalent undifferentiated SpA.

Meanwhile, PsA and RF-negative polyarthritis are too heterogenous and do not have an adult correspondence or are conditions exclusively seen in the pediatric population, such as early-onset ANA-positive JIA, previously included in both PSA and RF-negative polyarthritis [93]. Olygoarticular JIA and RF-negative JIA seem to be specific entities relevant to the pediatric age group [94].

Once these patients reach adulthood, they will be reclassified into another disease category, based on adult classification criteria that may confuse the patient, and they will also be directed to different treatment protocols corresponding to the new disease entity [95,96]. DCE-MRI may be a high-predictive value tool in clinically inactive JIA patients for disease activity on MRI. It is useful to detect subclinical synovitis in JIA by synovial hypertrophy with hyper-vascularization and quick inflow-outflow that should differentiate active disease patients from inactive ones with “normal” residual synovial thickening. Therefore, synovial maximum enhancement at the baseline on MRI is an important discriminating feature [97]. However, the risk of brain gadolinium deposits should be carefully considered before administering gadolinium to pediatric patients [98].

### 3.4. Other Diagnostic Tools

The synovial biopsy may help to understand and compare PsA biology to that of other chronic inflammatory joint diseases, such as RA and SpA, and also contribute to the development of a biological foundation for PsA classification, since the synovial membrane is one of the main tissues affected by PsA. Still, up until now, there have only been a few studies [99]. The pathogenesis of psoriatic spondylitis is fueled by increased synovial vascularity with high levels of osteoproliferative cytokines, such as bone morphogenic protein, as well as inflammatory cytokines such as TNF α, compared to uSpA and RA; lower levels of T cells were found compared to RA, without reaching statistical significance [100,101]. Some studies [99,102,103] have shown that PsA has a similar synovial phenotype with AS and undifferentiated SpA, unlike RA; in addition, further studies have shown that PsA has a histological particularity involving the synovial vascularity pattern, but newer studies have observed no histological statistical significant differences between uSpA and PsA [101,103]. The synovial biopsy may be an additional tool in the diagnosis of atypical patients and may guide the treatment response follow-up, improve the treatment decision-making, and accelerate decisions in phase I–II clinical trials [104], although further studies are required.

PsA diagnostic criteria include the presence of psoriatic skin lesions, but other skin diseases may either clinically or histologically mimic psoriasis, therefore it is necessary to differentiate it for a correct diagnosis. This group of conditions includes psoriasiform/psoriatic-like lesions, such as psoriasiform dermatitis—lichenified dermatitis, seborrheic dermatitis, pityriasis rubra pilaris, allergic dermatitis, atopic dermatitis, nummular dermatitis, lichen simplex chronicus, dermatophytosis, dyshidrotic eczema, psoriasiform drug rash, parapsoriasis, and less often, mycosis fungoides and secondary syphilis [101,105]. In these cases, the best diagnostic tool is a skin biopsy; clinical-histological concordance in the diagnosis of psoriasis was found to be significantly influenced by clinical features such as the typical scale and Auspitz’s sign, as well as histopathological evidence of suprapapillary thinning and granular layer absence. Lymphocytic exocytosis and the vertical orientation of collagen bundles were strongly linked with the diagnosis of psoriasiform dermatitis [106].

The lymphoid organs have not yet been investigated in PsA, despite the fact that activated immune cells can migrate from damaged psoriatic tissues to draining lymph nodes (LN), where they cause T helper cell differentiation and launch an inflammatory response. Since differentiated and activated T cells are essential for PsA, and LNs are the site of T-cell activation, this information will be extremely important for further studies [105,107,108,109].

### 3.5. Pharmacological Therapy

#### 3.5.1. Treatment Recommendations and Responses

Due to the lack of evidence of clinical trials of drug classes specific to axial PsA patients, most of the treatment guidelines are based on axial SpA guidelines [110].

Two groups, GRAPPA (Group for Research and Assessment of Psoriasis and Psoriatic Arthritis) and EULAR provide evidence-based treatment recommendations to providers. In the first instance, the gold standard is non-biological treatment—NSAIDs—for mild symptoms, but their action is limited, so in case of ineffectiveness, conventional synthetic disease-modifying antirheumatic drugs (csDMARDs) such as Methotrexate or Leflunomide are recommended, especially for peripheric disease with PsO. In case of a lack of effectiveness, biological disease-modifying antirheumatic drugs (bDMARDs) are recommended. Recent studies have shown that psoriasis patients treated with bDMARDs had a lower incidence of PsA [111].

Other studies have demonstrated the effectiveness for the treatment of active PsA of Janus kinase (JAK) inhibitors, such as tofacitinib, which is in phase II trials; filgotinib; upadacitinib, which is in phase II/III trials [110]; monoclonal antibody-like TNF inhibitors (entanercept, infliximab, adalimumab, golimumab, certolizumab); and interleukin antagonist agents (IL-17A—secukinumab, ixekizumab; and IL-12/23—ustekinumab, which are more efficient than they are in SpA, since both IL-17A and IL-23 are involved in PsA immunopathogenesis; however, IL-17A seems to be more centered on axial SpA, while IL-23 is more specific for the pathogenesis of inflammatory bowel disease) [112].

The target of the management of chronic inflammatory disease is the absence of both clinical and paraclinical manifestations [113]. For many clinicians it is still a burden to decide whether or not to start biological therapy in very early-disease PsA patients, but prospective exploratory studies have indicated that very early bDMARDs showed an important decline in skin symptoms [114]. TNF-α drugs are preferred over other biologic drugs [115] and further studies have analyzed depression or anxiety levels among these patients and demonstrated that TNF-α antagonist-treated patients showed an improvement in mental state [116,117,118].

The treat-to-target strategy used in other rheumatoid diseases has also been found to be effective in PsA patients, but in clinical practice it has not yet been implemented, due to the large variability of symptoms and missing uniform agreement on which target should be followed in PsA treatment. The target of the therapy is to achieve remission/inactive disease (DAPSA-REM) or, at least, minimal disease activity (MDA) which is a primary outcome instrument for clinical studies in PsA that has validity and a good correlation with other multi-domain measures [119]. Nevertheless, studies have shown that over 48% of all PsA patient groups underwent orthopedic surgery during the disease evolution, with a median age of around 51 years old, much earlier than the median age in the healthy population. The most common joint-sacrificing surgeries involved in axial and peripheral PsA have been knee prosthesis and hip prosthesis. It is important to note that 36% of these surgeries had been performed before the diagnosis, so the early detection and vigorous treatment of PsA from the start is mandatory to limit disease development, as it may lower the need for surgical treatments and offer a chance to enhance patients’ quality of life [120]. Inflammatory or mechanical lower back pain along with dactylitis is associated with a poor response to MTX, while TNF inhibitors and IL-17 inhibitor agents are strongly recommended as a first-line treatment as they both showed a good response in all six key domains of PsA (axial, peripheral, enthesitis, dactylitis, skin and nail lesions), as recommended at the last GRAPPA annual meeting of 2022 [118].

TNF inhibitors exceed the best treatment response in young patients with a short duration of disease, high C-reactive protein and normal body weight, and male sex is a good predictive factor. Meanwhile poor prognosis factors (dactylitis, high C-reactive protein, erosions) have better treatment responses to IL-23 inhibitor agents. Ustekinumab (IL12/23 inhibitor agent) has a similar efficiency in both normal and overweight patients, but it is indicated in mono/oligoarticular forms. Ustekinumab and apremilast (PDE inhibitor) obtained better responses in patients with low disease activity [118,121]. The treatment strategies for juvenile PsA are similar. JPsA patients more often present with peripheral arthritis and subsequently axial involvement, thus MTX is widely recommended as a second-line therapy, prior to Leflunomide or sulfasalazine, as it has better tolerability and fewer side effects [36].

Etanercept is the only TNF-inhibitor agent approved for pediatric use, recommended for children over 12 years old with a minimal response to conventional therapy. Ustekinumab (IL12/23) and scukinumab (IL-17A) are the subjects of ongoing trials. Tofacinib (JAK inhibitor) has been approved by the Food and Drug Administration and the European Medicines Agency for the treatment of early-onset jPsA in patients over 2 years old [10].

Methotrexate (MTX) is a widely used drug due to its effectiveness and the fact that it is less expensive than many of the alternatives recommended for the treatment of peripheral PSA and poly- and oligoarticular forms of JIA. Its long-term use has been related to severe liver toxicity, hepatic fibrosis, and eventually liver cirrhosis, although those findings only included old studies based on individuals who received a high dose administered daily. Newer studies have shown that MTX rarely induces liver fibrosis per se, but rather it acts synergistically with other hepatotoxic factors, such as diabetes mellitus or metabolic syndromes including obesity, increased body mass index, and dyslipidemia, which are strongly associated with PsA, or alcohol consumption [122].

Some studies have demonstrated the importance of the use of MRI imaging with low-frequency shear waves (range, 20–200 Hz) to create a visual elastogram that evaluates liver fibrosis (MRE). Most of the time, no data were available about the patients’ condition before the start of MTX treatment, so future studies are needed for pre-therapeutic follow-up and during treatment to obtain statistically relevant results. A recent Indian study [123] showed that MRE and serum liver biomarkers might be useful as a substitute for a liver biopsy, which is associated with sampling errors, procedural risks, poor repeatability, and increased morbidity. Moreover, MRE showed high diagnostic accuracy, with higher results compared to the Fibroscan and acoustic radiation force impulse (ARFI) methods and a positive correlation with biomarkers (AST/ALT ratio). MRE is substantially more accurate at detecting fibrosis in obese patients and is unaffected by BMI, ascites, or body habitus, unlike ultrasound-based quantitative elastography techniques [123]. Moreover, the cumulative dose and treatment duration did not correlate with MRE values and mean liver stiffness values [123,124]. However, MRE has its limitations in patients with hemochromatosis or hemosiderosis due to the high hepatic iron overload that will shorten T1, T2, and T2* relaxation times and lower the signal intensity [124].

#### 3.5.2. DMARDs Side Effects

As is the case for any medication, PsA drugs may have various side effects which can be exacerbated by each patient’s comorbidities.

csDMARD medications may be associated most often with an increased risk of liver toxicity; gastrointestinal symptoms, such as nausea, vomiting, diarrhea, abdominal pain, and bloating; neurological/psychiatric symptoms, including headaches, dizziness, depression, anxiety, and sleep disturbance; and a cutaneous rash, ulceration, nodules, and itching or blistering. Less often, patients encountered blood abnormalities such as thrombocytopenia, transaminase increase, anemia, alkaline phosphatase elevation, respiratory symptoms, fatigue, fertility problems, or alopecia [36,125].

bDMARDs may increase the risk of respiratory or vaginal infections, liver toxicity, skin infections or cancer, tuberculosis reactivation, congestive cardiac failure, and demyelination syndromes [126], while a recent study showed that over one third of patients discontinued their treatment for fertility or family planning reasons [36]. The study in question was performed between 1994 and 2019 to evaluate real-world treatment tolerability, and showed that csDMARDs are less tolerated than bDMARDs, with a mean duration of treatment until discontinuation up to 10 months, while the duration of treatment with biologic medication reached 18 months [36].

## 4. Discussion

Psoriatic arthritis is a highly prevalent disease and a relevant health issue. PsA patients represent a heterogeneous group, with high variability of non-specific symptoms and high overlapping with other pathologies of the SpA group, especially AS, which requires a multidisciplinary approach. 

The best tool for diagnosing PsA remains the CASPAR criteria. However, additional research is mandatory to create a worldwide accepted definition of axial involvement in PsA to develop diagnostic criteria that are easy to use, with high sensitivity and specificity. However, there are some psoriasiform skin lesions that may mimic psoriasis vulgaris which a skin biopsy might help to correctly diagnose. MRI has proved its essential role in early diagnosis, identifying pre-radiological modifications, such as bone marrow edema, fat metaplasia, and enthesitis. Asymmetrical sacroiliac erosions, cervical spine involvement, and skin psoriasis are more common characteristics of axial PsA than of axial SpA. MRI might give “false positive” results in cases of mechanical overload, such as for athletes or postpartum women, so clinical correlation remains essential for the patient’s diagnosis.

The analyzed studies generally agreed that axial PsA and AS have similar spinal activity and disability, with the findings that AS may have numerically more vertebra involved, and that axial PsA has a slightly higher susceptibility in the cervical spine.

One-third of patients may have SIJ involvement preclinically [17]. However, whole-spine MRI is not recommended for screening in the absence of symptomatology due to low confidence in inflammatory changes in the spine. Furthermore, synovial biopsies may be useful in PsA diagnosis to further our understanding of the disease pathology, and may be a cornerstone in the future development of new targeted therapies.

Further studies are necessary since patients generally consult several doctors until they are correctly diagnosed. On average, more than 7 years pass between the first medical visit and the diagnosis. Universally established criteria will allow better study comparisons, understanding of the disease, and treatment management.

The high overlapping characteristics of axial PsA and SpA have allowed medical professionals to diagnose and manage these patients, even without specific studies in PsA. Nevertheless, there are still limited data referring to axial PsA, and further studies are necessary to better understand the disease pathway, early signs, definition, and management.

Juvenile psoriatic arthritis is a self-standing entity, different from adult PsA, that is characterized by dactylitis, with female predominance in the early-onset disease, while the late-onset disease has more similar features to SpA. However, further revisions of the actual classifications, with less restrictive criteria, will allow for easier resettlement from juvenile to adult disease and more homogeneous groups.

Newer techniques such as MRE may help with the further follow-up of patients undergoing biological and non-biological treatments to evaluate treatment hepatotoxicity, as it showed a higher success rate and diagnostic accuracy than Fibroscan, with comparable results to liver biomarkers [122,123,124]. Further studies on a larger population are necessary to validate the actual data.

## Figures and Tables

**Figure 1 diagnostics-13-01342-f001:**
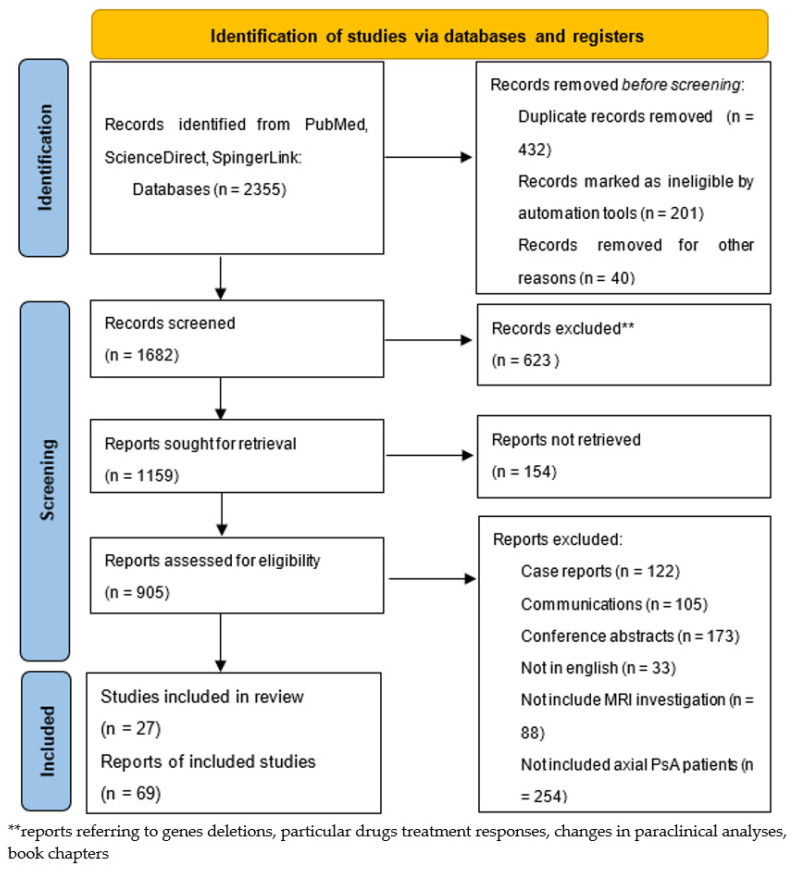
PRISMA flow chart of our study research.

**Figure 3 diagnostics-13-01342-f003:**
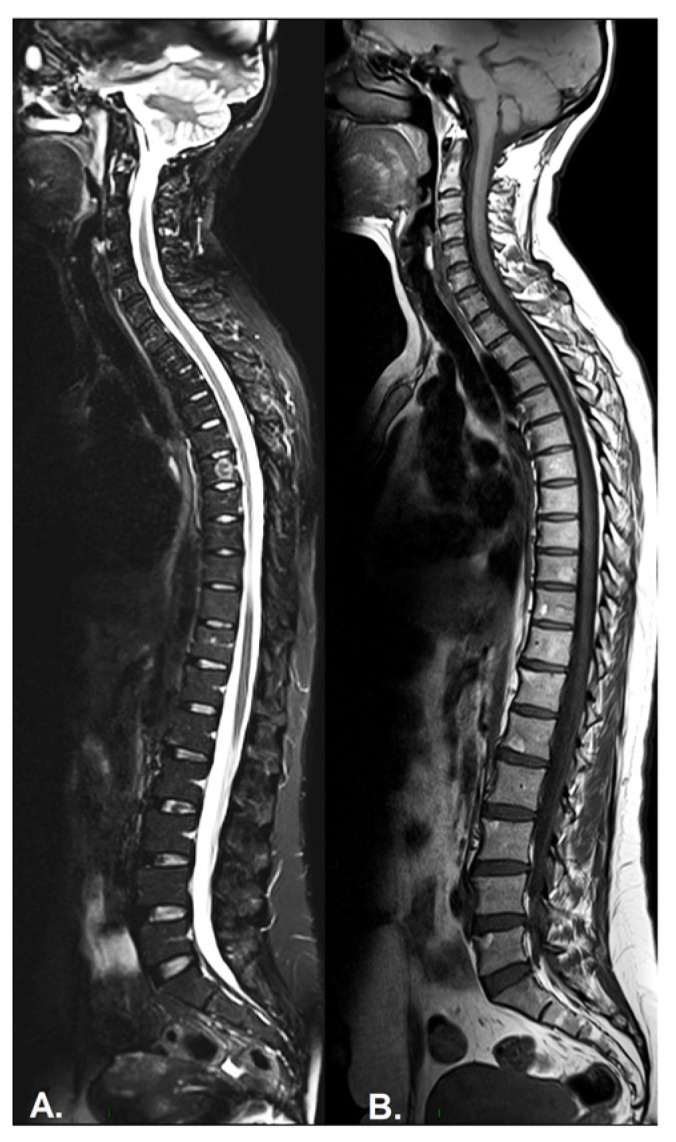
(**A**) Sagittal STIR, (**B**) T1-weighted sequence. Images show Romanus lesions as tiny subchondral bone marrow oedema (hyperintensities) in the antero-superior corners of lumbar vertebral bodies on STIR and, after the active phase, residual fatty transformation in the same regions on T1.

**Figure 4 diagnostics-13-01342-f004:**
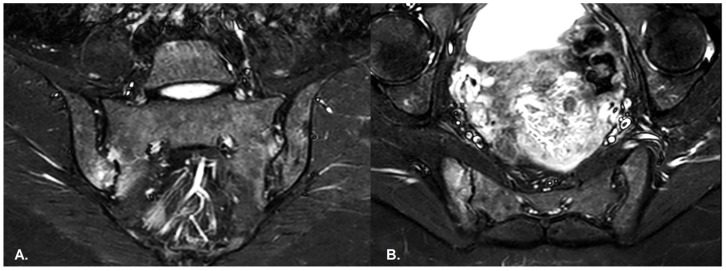
(**A**) Coronal and (**B**) axial STIR sequences show extensive subchondral oedema involving the sacroiliac joints, mainly on the right side and in the inferior regions, consistent with asymmetric active sacroiliitis in a patient with psoriasis.

**Figure 5 diagnostics-13-01342-f005:**
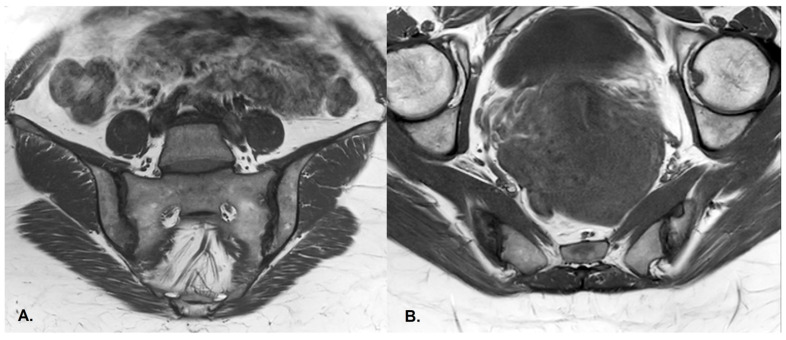
(**A**) Coronal and (**B**) axial T1-weighted sequences of the sacroiliac joints demonstrate subchondral sclerosis and erosions, predominantly on the right side, in the same patient as in Figure 4, consistent with sacroiliitis grade III.

**Table 1 diagnostics-13-01342-t001:** Most relevant selected articles cited in this review.

*Author*	Year	Study Population	Provided Information
*Braga et al. [17]*	2020	45 patients	MRI sacroiliitis evaluation
*Benavent et al. [18]*	2021	3684 patients	
*Feld et al. [19]*	2021	1354 patients	
*Furer et al. [20]*	2021	107 patients	
*Diaz et al. [21]*	2022	93 patients	
*Castillo-Gallego et al. [22]*	2013	76 patientes	Assement of inflammatory and structural damage
*Ye et al. [23]*	2019	121 patients	
*Poggenborg et al. [24]*	2015	48 patients	Comparison with other SpA entities
*Jadon et al. [25]*	2017	402 patients	
*Gensler et al. [26]*	2020	203 patients	
*Salinas et al. [27]*	2021	139 patients	
*Abdelaziz et al. [28]*	2021	100 patients	
*Arnbak et al. [29]*	2016	1037 patients	Inflammatory back pain prevalence, MRI findings, and clinical correlations
*Yap et al. [30]*	2018	171 patients	
*Aydin et al. [31]*	2017	1195 patients	
*Kivity et al. [32]*	2018	224 patients	
*Zisman et al. [33]*	2017	361 patients	Juvenile idiopatic arthritis, risks of psoriasis association, and other comorbidities
*Ekelund et al. [34]*	2017	440 patients	
*Brandon et al. [11]*	2018	26710 patients	
*Lamot et al. [35]*	2021	71 patients	
*Jawad et al. [36]*	2022	335 patients	Treatment side effects

**Table 3 diagnostics-13-01342-t003:** List of PsA classifications.

Classification	Details	Pros	Cons
**CASPAR**	Inflammatory arthritis of joint/spine/enthesis + ≥3 points from the following: Psoriasis of the scalp/skin - Current psoriasis, 2p - Personal history, 1p - Family history, 1p Nail dystrophy, 1p Negative rheumatoid factor, 1p Dactylitis, 1p - Present - Personal history Radiological evidence of juxta-articular syndesmophytes, 1p	- Most applied in clinical practice. - Easy to use. - High specificity (98%) and sensitivity (91%). - Also includes patients without psoriasis features or patients with a positive RF blood test. - Includes juxta-articular new bone formation as a feature of PsA, unlike other previous criteria. - Developed via multiple international cohort studies by PsA experts and now it has been widely adopted and implemented.	- The weakest aspect is the initial qualification criterion, since the inflammatory articular disease is not yet well-defined. - Does not include MRI findings, which are often more specific in early diagnosis.
**Moll and Wright**	Arthritis + Psoriasis + Negative rheumatoid factor. Divides PsA into five categories: - DIP joint only; - Asymmetrical oligoarthritis; - Polyarthritis; - Spondylitis; - Arthritis mutilans.	Historically, it was the simplest and most used criteria.	- Patients without skin psoriasis or those who have a positive RF blood test are not included. - Poor discrimination between PsA and RA. - Is no longer sustained over time and treatment.
**ESSG (European SpA Study Group)**	Synovitis/inflammatory back pain + Current psoriasis or personal history. Other features: - Arthritis; - Buttock pain; - Enthesitis; - Sacroiliitis; - Inflammatory bowel disease; - Episodes of acute diarrhea; - Urethritis [8].	- Easy applicability. - Allows for a diagnosis without evidence of skin psoriasis.	Lower sensitivity.
**Vasey and Espinoza**	Psoriasis/psoriatic nail dystrophy + Peripheral disease: - >4 weeks arthritis of DIP joint; - Asymmetrical peripheral arthritis (dactylitis); - Absent RF or rheumatoid nodule; - Radiographic changes (pencil-in-cup deformity, whittling of the terminal phalanges, fluffy periostitis, and bony ankylosis). Axial disease - >4 weeks of spinal pain and stiffness associated with motion restriction; - Symmetric sacroiliitis—grade 2; - Unilateral sacroiliitis—grade 3/4 (according to the New York criteria).	- Easy applicability. - Describes only two patterns of PsA. - Good specificity.	- Does not include patients without skin psoriasis or nail dystrophy. - Does not have enough statistical validation. - Lower sensitivity.
**ASAS**	Sacroiliitis in imaging studies (active inflammatory lesions in an MRI examination or X-ray changes defined according to the modified New York criteria) + ≥1 sign of spondyloarthropathy, or HLA-B27 antigen present + ≥2 signs of spondyloarthropathy SpA features: - Inflammatory back pain (IBP) - Arthritis - Enthesitis (heel) - Uveitis - Dactylitis - Psoriasis - Crohn’s disease/colitis - Good response to NSAIDs - Family history of SpA - HLA-B27 - Elevated CRP	- Easy to use. - High sensitivity (98.4%). - Includes MRI findings for the better assessment of early disease. - Developed via multiple international cohort studies by PsA experts and now it has been widely adopted and implemented.	Low specificity (77%) for PsA, as it has common features with other axSpA.
**ILAR** [47]	Definite jPsA. Arthritis and psoriasis or Arthritis with at least two of the following: - Dactylitis; - Nail pitting or onycholysis; - Psoriasis in a first-degree relative. Exclusion criteria: - Arthritis in an HLA-B27-positive male with arthritis onset after 6 years of age; - Ankylosing spondylitis, enthesitis-related arthritis, sacroiliitis with IBD, Reiter’s syndrome, or acute anterior uveitis in a first-degree relative; - Presence of the IgM rheumatoid factor on at least two occasions more than 3 months apart; - Presence of systemic JIA. Definite ERA. Arthritis and enthesitis, or arthritis or enthesitis with at least two of the following: - Presence or history of sacroiliac joint tenderness and/or inflammatory lumbosacral pain; - Presence of the HLA-B27 antigen; - Onset of arthritis in a male over 6 years of age; - Acute (symptomatic) anterior uveitis; - History of ankylosing spondylitis, enthesitis-related arthritis, sacroiliitis with inflammatory bowel disease, Reiter’s syndrome, or acute anterior uveitis in a first-degree relative. Exclusion criteria: - Psoriasis or a history of psoriasis in the patient or a first-degree relative; - Presence of an IgM rheumatoid factor on at least two occasions at least 3 months apart; - Presence of systemic JIA in the patient.	- Provides a single classification scheme accepted worldwide. - Better than previous criteria.	- Lower sensitivity and specificity compared to the CASPAR criteria. - Restrictive criteria, so patients risk being included in other subgroups. - Difficult to use in assessing patients that have psoriasis, or a psoriasis-like rash, or have relatives with psoriasis. - Classification has had some modifications over time, making the comparision of studies difficult.

**Table 4 diagnostics-13-01342-t004:** Comparison between the imaging features of axial PsA and AS.

	axPsA	AS
	44% HLA-B27-positive	90% HLA-B27-positive
**Cervical spine**	Facet joints fusion—more frequent	
**Dorsal spine**	Non-marginal syndesmophytes (spinal ligament origin) Asymmetrical Cranial to caudal evolution	Marginal syndesmophytes Symmetrical Caudal to cranial Paravertebral ossification
**Lumbar spine**		Facet joints fusion—more frequent
**Sacroiliac**	Less severe sacroiliitis Asymmetrical sacroiliitis Syndesmophytes extend into SIJ space Syndesmophytes may be observed in the absence of sacroiliitis	Severe sacroiliitis Symmetrical Syndesmophytes pass across the SIJ space
**BME**	Lower score in spine and SIJ	Higher score
**Fat metaplasia**	Less severe	More severe
**Erosions**	Less severe	More severe
**Peripheral involvement** **Enthesitis**	More often dactylitis, nail dystrophy Similar to AS	Less often Similar to PsA
**Uveitis**	Rare	Frequent
**Inflammatory bowel disease**	Similar prevalence	Similar prevalence

**Table 5 diagnostics-13-01342-t005:** Differential diagnosis of inflammatory from non-inflammatory back pain based on the typical location of BME.

Condition	Characteristic Location of BME
**axSpA**	Posterior lower quadrant of the ilium
	Dorsal-cartilaginous segment of the SIJ
**Healthy individuals**	Lower iliac bone
**Sports individuals**	Overlaps with axSpA
	Posterior lower quadrant of the ilium
	Anterior upper quadrant of the sacrum
**Postpartum**	Overlaps with axSpA
	Typically, no structural changes
**Degenerative**	Anterior and middle thirds
	Ligamentous segment of the SIJ
	Associated with the degeneration of the pubic symphysis

## Data Availability

Not applicable.

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
