# Peer review of "MRI Findings in Axial Psoriatic Spondylarthritis"

_diagnostics, 2023, doi:10.3390/diagnostics13071342_

Round 1
Reviewer 1 Report
The presented manuscript is a review article describing different aspects of MRI use in patients with PsA. Although this is an interesting topic which is not extensively elaborated in the literature, there are some major points which needs to be addressed.
1) The aim(s) of the review and the search strategy are not completely clear. More precisely, the methodology of literature review lacks appropriate PRISM flow chart which would explain the whole process.
2) The manuscript lacks tables with the most important papers summarised in one place. Moreover, it would also benefit from a table or figure showing the most important aspects of MRI findings in MRI patients.
3) It is not clear if the authors included studies with adult patients only, or they considered children as well. In any case, I believe the children should be included, and therefore ILAR criteria for PsA with all advantages and disadvantages explained and added to a table 1.
4) As mentioned in the manuscript, SpA and PsA are often associated with IBD and other similar manifestations. I belive this should be elaborated in more details. For example, I propose to include the following paper:
Lamot L, Miler M, Vukojević R, Vidović M, Lamot M, Trutin I, Gabaj NN, Harjaček M. The Increased Levels of Fecal Calprotectin in Children With Active Enthesitis Related Arthritis and MRI Signs of Sacroiliitis: The Results of a Single Center Cross-Sectional Exploratory Study in Juvenile Idiopathic Arthritis Patients. Front Med (Lausanne). 2021 Mar 8;8:650619. doi: 10.3389/fmed.2021.650619. PMID: 33763437; PMCID: PMC7982855.
Author Response
Point 1. The aim(s) of the review and the search strategy are not completely clear. More precisely, the methodology of literature review lacks appropriate PRISM flow chart which would explain the whole process.
Response 1. We now included PRISMA flow chart with our inclusion and exclusion reports strategy.
Point 2. The manuscript lacks tables with the most important papers summarised in one place. Moreover, it would also benefit from a table or figure showing the most important aspects of MRI findings in MRI patients.
Response 2. We added a table of the most important articles that are mentioned in review, classified by the main information provided.
Also, we created another table that includes MRI features in axial PsA and comparison with AS.
Point 3. It is not clear if the authors included studies with adult patients only, or they considered children as well. In any case, I believe the children should be included, and therefore ILAR criteria for PsA with all advantages and disadvantages explained and added to a table 1.
Response 3. We now included ILAR classification in our preexisting table of classifications. We described the main features in JIA, particularly in juvenile PsA.
Point 4. As mentioned in the manuscript, SpA and PsA are often associated with IBD and other similar manifestations. I belive this should be elaborated in more details.
Response 4. We created add more comparative findings and a comparison table between axial PsA and AS patients.
Reviewer 2 Report
Manuscript title: MRI findings in axial psoriatic spondylarthritis
The manuscript is well-written and presents interesting aspects of PsA and its imaging possibilities; however, a few things should be explained:
1. The authors discuss the role of MRI and various aspects of MRI findings, but there is no MRI imaging showing the specific findings
2. The authors present various criteria. Which of them are most often used in clinical practice and why?
3. In the whole work, there is only one table. A graph summarising the consecutive steps in pharmacologic therapy is needed.
4. Is the mentioned therapy of PsA safe? The side effects of discussed DMARD should be noted. Are there some DMARD preferences in different age-group?
Author Response
Point 1.The authors discuss the role of MRI and various aspects of MRI findings, but there is no MRI imaging showing the specific findings
Response 1. We now included some relevant MRI pictures.
Point 2. The authors present various criteria. Which of them are most often used in clinical practice and why?
Response 2. We evidenced that CASPAR is the most clinical used criteria in PsA patients and also highlighted the strengths and weaknesses compared to other previous classifications.
Point 3. In the whole work, there is only one table. A graph summarising the consecutive steps in pharmacologic therapy is needed.
Response 3. We created several new tables, including pharmacological treatment based on the latest GRAPPA guidelines.
Point 4. Is the mentioned therapy of PsA safe? The side effects of discussed DMARD should be noted. Are there some DMARD preferences in different age-group?
Response 4. We included side effects for each treatment class, including aproved treament for pediatric population and age recommandations.
Round 2
Reviewer 1 Report
The authors have successfully addressed all of the concerns raised during the review process.